# Impact of Nitric Oxide on Polymorphonuclear Neutrophils’ Function

**DOI:** 10.3390/biomedicines12102353

**Published:** 2024-10-16

**Authors:** Richard Kraus, Elena Maier, Michael Gruber, Sigrid Wittmann

**Affiliations:** Department of Anaesthesiology, University Hospital Regensburg, Franz-Josef-Strauss-Allee 11, 93053 Regensburg, Germany

**Keywords:** PMN function, nitric oxide, migration, surface epitopes

## Abstract

Background: There is increasing evidence that nitric oxide (nitrogen monoxide, NO) significantly influences immune cellular responses, including those from polymorphonuclear leukocytes (PMNs). Objective: The aim of this study was to examine a possible effect of NO on PMNs’ function (chemotaxis, production of reactive oxygen species (ROS), and NETosis) using live cell imaging. Moreover, we investigated PMN surface epitope and neutrophil oxidative burst under the influence of NO by flow cytometric analysis. Methods: Whole blood samples were obtained from healthy volunteers, and PMNs were isolated by density centrifugation. Live cell imaging using type I collagen matrix in µSlide IBIDI chemotaxis chambers was conducted in order to observe N-formyl-L-methionyl-L-leucyl-phenylalanine (fMLP)-stimulated PMN chemotaxis, ROS production, and NETosis. In the test group, NO was continuously redirected into the climate chamber of the microscope, so the chemotaxis chambers were surrounded by NO. The same experimental setup without NO served as a control. In addition, isolated PMNs were incubated with nitrogen monoxide (NO) or without (the control). Subsequently, flow cytometry was used to analyze neutrophil antigen expression and oxidative burst. Results: Our live cell imaging results demonstrated a migration-promoting effect of NO on PMNs. We observed that in the case of prior stimulation by fMLP, NO has no effect on the time course of neutrophil ROS production and NET release. However, flow cytometric analyses demonstrated an increase in ROS production after pretreatment with NO. No NO-dependent differences for the expression of CD11b, CD62L, or CD66b could be observed. Conclusions: We were able to demonstrate a distinct effect of NO on PMNs’ function. The complex interaction between NO and PMNs remains a major research focus, as the exact mechanisms and additional influencing factors remain elusive. Future studies should explore how varying NO concentrations and the timing of NO exposure relative to PMN activation affect its influence.

## 1. Introduction

Being a particularly reactive biomolecule, NO is of multifaceted importance in the human organism. Due to the lipophilic properties of NO, it is able to diffuse through membranes and thus acts beyond cell boundaries. In the vascular system, NO acts as a messenger and influences the vascular tone, resulting in localized vasodilation. In addition, it inhibits both thrombocyte and leukocyte adhesion to the vascular walls. NO is produced by NO synthases. The short half-life of NO of approximately five seconds within the human body (roughly comparable to an aqueous solution) requires immediate NO synthase activation when needed [1,2,3]. Related to this, a physical proximity between NO production and the target is needed. Accordingly, several isoforms of NO synthase exist [4].

Endothelial NO synthase (eNOS) is of significant importance in the regulation of the vascular system and, thus, in blood pressure control. During vasodilatation, the NO synthesized in the endothelial cells reaches the smooth vascular muscle cells via diffusion [3]. Within the coagulation system, NO acts via inhibition of thrombocyte adhesion and aggregation to the vascular wall [5,6]. Furthermore, NO reduces the expression of adhesion molecules for the diapedesis of leukocytes [7].

Besides signal transduction, NO has an important function in the defense against pathogens. Similar to ROS, it has a cytotoxic effect in high doses and is therefore released by PMNs [5,6]. After activation, the responsible enzyme, the inducible NO synthase (iNOS), continuously releases NO in large quantities. Being a free radical, NO has an antipathogenic effect itself. It is also partially converted into highly microbicidal peroxynitrite by reaction with superoxide [8].

The most commonly used NO-releasing pharmaceutical is nitroglycerin. Other NO-donors are isosorbide mononitrate and isosorbide dinitrate, which can also be used in the treatment of angina pectoris [9]. In everyday clinical practice, the vascular effect of NO is mainly used for therapeutic purposes in cardiovascular diseases. Nitroglycerine spray is used as an emergency medication to treat acute coronary syndrome, acute left heart insufficiency, and catheter-induced coronary spasms [10], whereas molsidomin is used prophylactically in case of angina pectoris complaints [11]. With these medications, the therapeutically effective NO is produced secondarily in the body and not supplied directly.

In intensive care medicine, NO can also be administered as a gaseous drug. This is, for example, indicated for newborns with hypoxic respiratory insufficiency, as well as for peri- and postoperative pulmonary hypertension. In the process, concentrations between 5 and 40 parts per million (ppm) are administered by inhalation [12].

Furthermore, NO has been classically described as an anti-inflammatory molecule that inhibits leukocyte adhesion [7]. Conversely, this could mean a potential inhibition of the adhesion and migration capacity of PMNs.

PMNs, with 50–70% of all circulating leukocytes, are the most mobile and numerically largest cellular component of the innate immune system and an important first line of defense within the innate immune response. PMNs are formed in the bone marrow, released into the blood, and reach their site of action via chemotaxis [13]. For this purpose, PMNs can recognize extracellular chemotactic concentration gradients and move towards higher concentrations based on the gradients. PMNs have receptors for chemokines and chemoattractants, such as the endogenous molecules complement factor 5a (C-X-C motif), ligand 8 (CXCL-8), and leukotriene B4 (LTB4) released during an inflammatory response, but also for exogenous molecules, such as the peptide N formyl-methionyl-leucyl-phenylalanine (fMLP) released by bacteria [14]. Once at the site of action, PMNs utilize the phagocytosis of pathogens as well as the production of reactive oxygen species (ROS). During the so-called “oxidative burst reaction”, phagocytosing PMNs exhibit a significant increase in their oxygen consumption, which is driven by the NADPH-dependent production of superoxide anions (O_2_^−^). These superoxide anions trigger the production of ROS [15]. Moreover, at the end of a cytolytic process, PMNs release their nucleus into the extracellular space as a net-like DNA structure known as neutrophil extracellular traps (NETs). These NETs contain histones and cationic peptides on their surface. Once released, they bind to bacteria and fungi, immobilizing and ultimately killing them. This phenomenon, primarily occurring at sites of inflammation, is referred to as NETosis [16].

In 2010, Patel et al. examined the influence of NO donators on NETosis. In this context, NO induced an increased production of free radicals, which, in turn, stimulated NETosis [17]. Furthermore, Manda-Handzlik et al. described a more direct correlation between NETosis and RNS (NO and peroxynitrite) [18]. In the process, peroxynitrite and NO directly seem to have an inducing effect on NETosis. S-nitroso-N-acetylpenicillamine (SNAP) served as an NO donor in the experiments [18]. In both studies [17,18], NADPH oxidase and myeloperoxidase (MPO) were essential for NET activation by NO. However, neither study examined whether NO also leads to an accelerated occurrence of PMN NETosis.

Clancy et al. were able to demonstrate that NO has an effect on oxidative burst. NO had an inhibiting effect on NADPH oxidase, which resulted in reduced superoxide production. However, only an influence on the enzyme during incubation with NO prior to its stimulation with N-formyl-L-methionyl-L-leucyl-phenylalanine (fMLP) or Phorbol-12-myristat-13-acetat (PMA), i.e., before the aggregation of membrane-bound and cytosolic components of NADPH oxidase, was described [19]. Inter alia, NO is released in inflammatory processes of PMNs during nitrosative stress [20].

NO can interact with the superoxide anion (O_2_^−^) to form toxic peroxynitrite, leading to lipid peroxidation, which indicates that lipid peroxidation can occur in the absence of iron through a peroxynitrite-mediated mechanism and suggests that NO may act as an antioxidant when produced in large amounts [21]. Moreover, physical exercise activates the NO-cGMP (cyclic guanosine monophosphate) signaling pathway in human neutrophils, leading to increased expression of myeloid cell leukemia 1 (Mcl-1) and a delay in apoptosis. This suggests that exercise may help prolong the lifespan of neutrophils by enhancing the iNOS-NO-cGMP-Mcl-1 pathway [22].

After reviewing the current studies, it has become evident that NO has an impact on the cellular reactions of PMNs. An aspect that had gone unnoticed so far was the time component of the different PMN functions. Therefore, in our study, live cell imaging was used to quantify the influence of NO on migration, ROS production, and NETosis of PMNs. In addition, flow cytometry was conducted to examine the surface epitopes CD11b, CD62L, and CD66b (CD = cluster of differentiation), as well as the oxidative burst.

## 2. Materials and Methods

### 2.1. Vote of the Local Ethics Committee

The study was conducted in accordance with the principles of the Declaration of Helsinki and was approved by the local ethics committee of the University of Regensburg (file number: 16-101-0322).

### 2.2. Sample Collection and Isolation of PMNs

After the informed consent of the volunteers, whole blood samples were taken from an antecubital vein with a lithium heparin-anticoagulated blood collection set (all blood collection materials were from SARSTEDT AG & Co. KG, Nuembrecht, Germany). A total of *n* = 34 blood samples were collected. There were 13 samples from male volunteers and 21 samples from female volunteers. The age range of the participants was between 22 and 54 years, and all of them were healthy at the time of blood collection. PMNs were isolated by leuco/lymphospin density gradient centrifugation and subsequently resuspended in Roswell Park Memorial Institute (RPMI)-1640 culture medium (Pan Biotech Ltd., Aidenbach, Germany) with 10% fetal calf serum (Sigma-Aldrich, Taufkirchen, Germany) at a concentration of 18 × 10^6^ cells/mL, in accordance with the protocol described by Doblinger et al. [23].

### 2.3. Preparation of the Chemotaxis Chambers

For the chemotaxis experiments, μSlide chemotaxis chambers^®^ (IBIDIIBIDI GmbH, Martinsried, Germany) were utilized (see Figure 1). These chambers feature a central channel at each of the three positions, which was filled with a type I collagen matrix in a liquid state mixed with the isolated PMNs. After 30 min hardening time at 37 °C, 5% CO_2_, and 50% air humidity, the left reservoir adjacent to the channel was filled with a 10 nM fMLP solution (Sigma Aldrich, Taufkirchen, Germany), while the right reservoir received RPMI-1640 medium.

### 2.4. Live Cell Imaging

The IBIDI Heating System and IBIDI Gas incubation System for CO_2_ created a controlled environment with 37 °C, 5% CO_2,_ and 50% air humidity for the subsequent live-cell imaging. Depending on testing with or without NO, the climatic chamber was set to the corresponding ambient conditions. In order to influence PMNs, NO was continuously redirected into the climate chamber of the microscope. For this purpose, NO was mixed in N_2_ with 21% partial O_2_ pressure using the anesthesia machine Trajan 808 (Draeger, Luebeck, Germany). For the control tests, the climate chamber was ventilated with compressed air instead.

Following the method outlined by Doblinger et al., migration and fluorescence were evaluated using a DMi8 inverted microscope (Leica Microsystems, Wetzlar, Germany), with photographic images taken using a DFC9000 GT SCMOS black-and-white camera (Leica Microsystems). The control of the microscope and the observation process were computer-assisted by the Application Suite X (Version 10) software platform (Leica Microsystems) [23]. The total observation time of each experiment under the microscope was 6 h. Four images at 100× magnification per IBIDI channel were acquired at intervals of 30 s in each cycle, leading to the recording of the processes of each IBIDI channel in 720 individual images. ROS production was measured using 1 μM dihydrorhodamine 123 (DHR; Thermo Fisher Scientific, Waltham, MA, USA). NETosis was observed through the use of 5 μM 4′,6-diamidino-2-phenylindole (Sigma-Aldrich), as described in an earlier publication of our research group [23,24].

### 2.5. Evaluation of the Microscope Images Showing Migration

The image series produced by the microscope was examined with the Imaris^®^ 9.0.2 software (Bitplane AG, Zuerich, Switzerland). To measure migration, axes in the x and y directions were established for each channel, similar to a Cartesian coordinate system, across the entire length of the image area under consideration, enabling the determination of specific migration quantities (see Figure 2). Parameter TrackLength is the final distance that a PMN cell has migrated in the matrix. Track Displacement X or Y describes the distance that the PMN cell has traveled on the X or Y axis. TrackStraightness can be used to determine the extent to which the PMNs migrate directly in the matrix. The value is calculated by dividing the distance migrated (TrackDisplacementLength) by the definite distance migrated (TrackLength). For comparison of the neutrophil migratory behavior, the whole observation period was divided into time slots of 30 min each. The filters TrackLength > 25 µm and TrackDuration > 900 s were set.

### 2.6. Evaluation of the Microscope Images Showing Immune Effects

All dyes utilized for measuring ROS and NETosis appeared as colored areas on the fluorescence images in Imaris^®^ 9.0.2 software (Bitplane AG). The software automatically recognized these areas in each image and exported the data to a separate Excel file detailing the surface areas [μm^2^] of each color in every image. The combined areas at the same time point were then summarized in Excel. ROS production detection using DHR 123 showed a parabolic curve (see Figure 3). The time taken to reach maximum ROS production, TmaxROS, was determined using a third-degree polynomial trendline from the relevant section of the graph. The first derivative of this formula and its zero point were calculated.

NETosis determination with (4′,6-diamidino-2-phenylindole) DAPI staining resulted in sigmoidal fluorescent curves (See Figure 4). The time to achieve the half-maximum effect, Et_50_, was calculated using Phoenix NLME (Certara L.P., Radnor, Pennsylvania).

### 2.7. Flow Cytometer Experiments

#### 2.7.1. Preparing the ROS Measurement Series

Figure 5 gives an overview of the experimental procedure for the tube preparation for oxidative burst measurement with flow cytometry.

In order to bring PMNs into contact with NO, a method in which phosphate-buffered saline (PBS) was saturated with NO using a gas washing bottle was established. This buffer, blended with NO, was mixed with the isolated PMNs (see Section 2.2). Untreated PBS was used for the control samples. The samples were incubated for a total of 30 min at 37 °C to measure the ROS activity. PMNs with NO influence were placed in a heating chamber, which was continuously fumigated with NO. For this purpose, a gas cylinder with 200 vol ppm NO in N_2_ was used.

In order to obtain the real intensity of stimulated ROS production, PMNs´ rhodamine fluorescence was measured without stimulus as a basic value. This allowed the difference to be calculated later. PMNs were stimulated with fMLP or with PMA, which were used as a positive control.

There were, therefore, 12 tubes for flow cytometry per experiment prepared, with 6 samples from the control group and 6 samples with NO as an influencing factor. Duplicates were measured in each case. During PMN isolation, 10 µL each of the two dyes, DHR (10 μM) and seminaphtharhodafluor (SNARF) fluorescent dye (10 μM, Invitrogen (Thermo Fisher, Waltham, MA, USA), were initially added to all tubes. In addition, 10 µL of TNFα (1 μg/mL, PeproTech Inc. ((Thermo Fisher, Waltham, MA, USA)) was pipetted into the fMLP assays. While the PBS could also be added to the control samples, the NO-fumigated PBS was only added to the tubes shortly before the PMN concentrate was added in order to ensure optimum saturation. After all tubes were filled with 1 mL of the corresponding PBS, 20 µL of the PMN concentrate of the 5 different subjects could be added to the assigned tubes.

All samples were then incubated at 37 °C in a water bath, the control group in room air, and the NO group in a chamber continuously fumigated with NO. After the first incubation, 10 µL PMA (10 µM) was pipetted into all positive controls and 10 µL fMLP (10 μM) into the fMLP tubes. The samples were then incubated again for a further 20 min under the conditions described above. In order to determine the extent of the burst reaction at the same time, all tubes were placed in a refrigerator at 4 °C for 5 min immediately after the end of the 20 min incubation.

Due to the abrupt drop in temperature, the ROS reaction was stopped, and thus, the amount of fluorescent rhodamine-123 stagnated. Shortly before the measurement, 10 µL of propidium iodide (PI) was pipetted into each tube to identify the dead cells. The substrate marks exposed DNA. Table 1 shows the composition of the different sample preparations for flow cytometric respiratory burst measurement.

#### 2.7.2. Preparation of the Antigen Test Series

After PMN isolation (see Section 2.2), the preparation of the measuring tubes with the 3 antibodies against surface proteins (CD11b-PE, CD62L-FITC, and CD66b-APC (Biolegend, San Diego, CA, USA)) was processed parallel to Section 2.7.1. Thereby, 5 µL of each of the 3 different antibodies were added to the tubes for measuring surface protein expression. In the end, there were 8 tubes per test subject, whereby duplicates were measured in each case. Table 2 shows the composition and pretreatment of the tubes for flow cytometric measuring antibody expression.

#### 2.7.3. Measurement and Examination Flow Cytometry Data

Neutrophil respiratory burst and cell-surface antigen expression levels were assessed using flow cytometry with a FACSCalibur flow cytometer and CellQuest Pro software Version 5.2 (both from BD Biosciences, San Jose, CA, USA). The data obtained was analyzed with FlowJo Version 10.0.7 (FlowJo LLC, Ashland, OR, USA), as previously described (see Figure 6). Ten thousand cells were counted per tube, and the corresponding values were recorded.

### 2.8. Statistics

The statistical analyses were carried out using SPSS 27 (IBM). After testing for normal distribution using the Shapiro–Wilk test, a *t*-test with independent samples was performed for normally distributed variables and the non-parametric Mann–Whitney U test for non-normally distributed variables. PMN functionalities under the influence of NO were compared with those from the control experiments. Thus, a dichotomous variable (with NO vs. without NO) was examined for statistical differences, which means that multiple comparisons or correlations did not occur. An error probability of *p* < 0.05 was considered statistically significant.

## 3. Results

### 3.1. Results of Neutrophil Migration

#### Results of Parameter TrackLength

In the statistical calculations, *n* = 14,343 tracks of the control group (air) and *n* = 23,749 tracks of PMNs fumigated with NO could be included. Figure 7 shows the neutrophil migratory behavior. Our results showed that in both groups, the determined track lengths diminished with increasing duration of the experiment.

There was a significant increase in TrackLength of the NO-fumigated PMNs (see Figure 7). As can be recognized in Figure 7, TrackLengths between the two test groups differed most significantly in the first two series of starting images 1 (first 30 min) and 61 (second 30 min). Consequently, the track lengths in these time windows were investigated separately for statistically significant differences (see Table 3).

### 3.2. Immune Effects in Live Cell Imaging

In our experiments, NO did not significantly affect neutrophil ROS production (see Table 4).

### 3.3. Result of Flow Cytometry

#### 3.3.1. Quantification of the Oxidative Burst Reaction

After stimulation with fMLP/TNFα, the mean values of the rhodamine-123 measurements (*n* = 19) were significantly increased in the NO group versus the control group (see Figure 8a). No differences were found after stimulation with PMA (*n* = 17; see Figure 8b and Appendix A).

#### 3.3.2. Analyses of Neutrophil Surface Epitopes

To investigate whether NO exerts an influence on neutrophil surface epitopes, the expression of CD11b, CD62L, and CD66b was examined under NO influence. These surface epitopes are important for chemotactic transmigration through blood vessel walls (CD11b and CD66b) and subsequent pathogen defense by phagocytosis (CD66b). Increased CD11b and CD66b, and decreased CD62L, would indicate PMN activation.

In the statistical analysis of the selected surface proteins, comparative analyses of *n* = 11 samples without and with NO fumigation were performed. The flow cytometric results revealed no differences for the expression of CD11b (*p* = 0.675), CD62L (*p* = 0.741), or CD66b (*p* = 0.555; for details see Appendix A).

## 4. Discussion

As pointed out in the introduction, the role of NO on the human organism is controversially debated. In this study, the direct impact of gaseous nitric oxide on PMNs´migration ability, oxidative burst, and NETosis was investigated. The experiments aimed to compare the timing parameters of the oxidative burst reaction and NET formation under the influence of NO. Additionally, the distance migrated by PMNs was examined to determine any potential effect of NO on their migration ability. Furthermore, this study assessed the extent of PMN ROS production and PMN surface epitope expression in the presence of NO. Our data show a distinct effect of NO on PMN function. A classification and interpretation of the results is provided below.

### 4.1. Influence of NO on Oxidative Metabolism

Various studies were able to demonstrate that high NO concentrations (>1 µM) can lead to oxidative and nitrosative stress [25,26,27]. On the other hand, the protective effects of NO were also observed under oxidative and nitrosative stress [28]. The fact that NO seems to have an important influence on the human organism, above all in inflamed tissue, has already been examined by several studies [29,30,31].

The flow cytometric tests of our project showed a significant increase in ROS with NO treatment and fMLP stimulation. These results are contrary to the observations of Clancy et al. However, the differences in the experimental setup must be taken into account. Clancy et al. conducted testing with NO influence before and after stimulation with arachidonic acid. Their results showed that NO caused a significant inhibition of NADPH oxidase-dependent ROS production when incubated for 5 min with NO before the activation. In our study, PMNs were stimulated with fMLP already before the NO influence. However, no effect was observed in the experiments of Clancy et al. with simultaneous administration of NO and arachidonic acid. This indicates that NO only inhibits NADPH oxidase when membrane-bound and cytosolic components of the NADPH oxidase are still present separately in an unstimulated state [19]. According to the results of this study, NO leads to increased ROS production with already activated PMNs.

Another study examined the influence of NO on the ROS production of PMNs. In the process, Clancy et al. were able to demonstrate that NO inhibits the production of ROS by inhibiting NADPH oxidase. ROS release was measured on vital isolated PMNs. In the process, there was a decrease in superoxide release with prior incubation in NO-saturated buffer. For a more detailed examination, the cell-free NADPH oxidase was additionally influenced by NO. After that, the concentration of oxygen radicals was measured again methodically, and superoxide anion generation was monitored by the determination of the reduction of cytochrome C in the presence or absence of superoxide dismutase [19].

Some research groups and authors also postulate a proinflammatory effect of NO [29,32,33]. Above all, large quantities of NO can potentiate an inflammatory reaction. In this context, it must be taken into account that the effect of NO depends on various factors. Of particular importance are cellular context, time of NO influence, and previous priming of the cells to be examined [29]. Also, NO concentration seems to be of particular relevance since several studies observed a concentration-dependent inhibition and stimulation of the reactive burst of PMNs [34,35,36]. This complicates the comparability of the study results with different analysis methodologies, stimuli, and concentrations. Moreover, the activity of PMNs may also be only affected by the isolation procedure. Density gradient centrifugation, for example, activates the PMNs [37].

Nevertheless, the results of our flow cytometric tests support the thesis of a proinflammatory effect of NO in the sense of increased ROS production by PMNs. In this context, several previous studies were able to demonstrate ROS-induced ROS release (RIRR) [38,39,40]. As a subtype of RNS, NO is closely associated with ROS with regard to pathophysiological processes. Besides ROS, NO is also released by stimulation of iNOS during the activation of PMNs [5,16,39]. Conversely, ROS are also produced after the activation of NOS [41]. Both radical groups have an oxidizing effect. In the sense of a positive feedback loop, this represents an explanatory approach for the increased ROS release in the flow cytometric tests conducted here. As with RIRR, NO could lead to increased ROS and RNS release by PMNs. Such a correlation between NO and oxidative stress was described in an article by Kuklinski [32]. Also, Wink et al. described NO as an indirect mediator for nitrosative and oxidative stress [42], which could, inter alia, be explained by the stimulation of the reactive burst of PMNs. Congruent to the findings obtained here, a study by Manda-Handzlik et al. observed an increase in ROS production under the influence of NO. In their study, the oxidative burst was assessed by flow cytometry dihydrorhodamine (DHR) 123 oxidation assay and nitroblue tetrazolium (NBT) assay [18].

Since the influence of NO on the quantity of ROS production has already been examined [19], further variables are of interest at this point. Our study was able to additionally evaluate the time aspect of the reactive burst by means of the microscope test series. The results do not show any significant effects of NO on the time course of ROS production. Thus, it can be assumed that microbial control by oxygen radicals is neither delayed nor accelerated under the influence of NO. This is, inter alia, relevant because iNOS with consecutive NO release also occurs in an inflammatory milieu [20,43].

The effect of NO on inflammatory processes also plays a role with regard to therapeutically used NO inhalation, e.g., in case of lung failure [44]. The affected patients are often multimorbid and susceptible to infections. The inhibition of pathogen control by PMNs would be a crucial disadvantage and would have to be included in the risk stratification for NO inhalation. However, the results of our live cell imaging showed no significant influence of NO on the time component of neutrophil ROS production. On the contrary, the results of our flow cytometric tests indicated an increased production of ROS. In a clinical setting, this could lead to an increase in oxidative stress in NO-inhaling patients in the context of inflammatory processes. Since, inter alia, cardiothoracic surgery patients are also treated with inhalative NO in the context of pulmonary hypertension, and the inflammatory activity is, in general, already increased after an operation, the ROS-promoting effect of NO should be taken into account and weighed against the benefits of such treatment [12,45].

### 4.2. Influence of NO on NETosis

Concerning the question of whether NO influences NETosis, very few studies have been published so far. By means of the microscope tests of this study, the time of maximum increase in NETosis activity could be determined via fluorescence measurements. The results showed no difference regarding the time parameters of NETosis during NO influence with prior activation of PMNs.

If these facts are compared with the results for ROS production under NO influence, the findings are congruent. The time course of ROS production in our live cell imaging experiments was not influenced by NO. This is understandable since the formation of NETs is significantly initiated by ROS [46]. RNS are released during PMN oxidative burst, too, and may have similar physiological effects to ROS. Therefore, it is theoretically possible that RNS (including NO) have an activating effect on PMN NETosis. Also, Patel et al. examined the effect of NO on the NETosis of PMNs. They stimulated PMNs with NO donors sodium nitroprusside (SNP) and S-nitroso-N-acetylpenicillamine (SNAP), which indirectly led to the initiation of NETosis. The underlying mechanism was NO-mediated ROS production. Thus, the reactive oxygen radicals ultimately led to the formation of NETs [17]. An essential difference to the present study was the absence of prior PMN activation by a chemoattractant, such as fMLP. Thus, it can be assumed that NETosis was ultimately activated by ROS in both studies. While Patel et al. directly stimulated PMNs with NO to produce ROS, this study used fMLP as an activator of ROS production. Presumably, the ROS ultimately led to NETosis in both test series. Thus, Patel et al. were able to demonstrate the initiation of NETosis by NO donors. Another study by Manda-Handzlik et al. also examined the effect of RNS and NO on PMN NETosis. In line with the results of Patel et al., a stimulating effect of NO on NETosis could be demonstrated. This effect was similarly based on ROS-dependent induction of NETosis [17]. In addition, Manda-Handzlik et al. were able to show not only a stimulating but also a potentiating effect with already activated PMNs. However, the potentiating effect depended on the stimulus used. Tested were, among others, PMA, TNF-alpha, PAF, IL-8, and calcium ionophore (CI). Only with PMA and CI could further potentiation by NO be achieved. fMLP was not used in this study [18]. Meanwhile, the experimental setup used here showed that NO no longer leads to an accelerated release of NETs with fMLP-stimulated PMNs.

### 4.3. Influence of NO on PMN Surfaces

In our study, the PMN surface epitope expression (CD11b, CD62L, and CD66b) remained unchanged. However, various studies pointed out that NO has a significant influence on the adhesion of PMNs to the microvascular endothelium.

A study by Hickey et al. examined leukocyte recruitment in iNOS-deficient mice and was able to demonstrate increased leukocyte adherence to the microvascular vessels [47]. Kubes et al. described a clear increase in leukocyte adhesion to the mesenteric veins of cats after inhibition of eNOS [48]. Also, Suematsu et al. examined in vivo mechanisms leading to the suppression of leukocyte activation in the mesenteric veins of rats after inhibition of endothelial NO production. However, this effect could, inter alia, be suppressed by pretreatment with antibodies against CD18 on the PMN surface [49]. A similar study by Akimitsu et al. also observed increased leukocyte adhesion in venules of rats after inhibition of NO synthase. Also, here, the effect of NOS inhibition could be neutralized by means of CD18 antibodies. This fact, in turn, suggests that a reduced NO concentration leads to an increase in leukocyte activation via increased expression of the above-mentioned surface proteins. Conversely, a reduced expression of CD18 due to increased NO concentrations would have to be assumed. Since CD18 and CD11b are components of a Mac-1 integrin [50], this assumption should also be transferable to CD11b, which was examined by flow cytometry in this study. However, this study was not able to confirm an inhibition of CD11b expression on the PMN surface. No differences with regard to CD11b antibody fluorescence could be detected after NO stimulation. In addition, CD62L and CD66b expression did not show any significant change. However, the relevant difference in study design must be taken into account. While the cited studies had examined leukodiapedesis in vivo [47,48,49,51], our study was able to evaluate the expression of the different surface proteins after NO stimulation in vitro in isolation. With regard to our present study, the effect of NOS inhibitors on leukocyte recruitment seems to be based on a context that is more complex than the direct influence of NO concentration on the expression of said surface proteins.

### 4.4. Choice of NO Concentration in Buffer Solutions: Discussion and Rationale

Inhibitory effects of NO on PMNs were postulated [19,48], but also stimulatory effects could be identified [17,18,38]. In this context, it is important to differentiate between NO concentrations. Different concentrations exert various physiological effects on tissues. While low concentrations (NO < 1–30 nM) already have an effect on signal transduction, high NO concentrations (NO > 1 µM) are released by macrophages through activation of iNOS in the fight against pathogens and tumors [25,52]. PMNs, which are also part of the non-specific immune response, are thus surrounded by NO in an inflammatory environment.

In our study, PMNs were also exposed to high NO concentrations. In view of the method used by us, concentrations about 160 ppm in the gas phase around the chemotaxis chambers have to be assumed (200 ppm in N_2_, mixed with 21 vol% O_2_). According to Henry´s law, the concentration of NO in the aqueous phase was, therefore, about 3.0 nM [53,54]. Due to the small size of the NO molecule and the good diffusion properties [55], the gas was able to diffuse into the gel matrix with the PMNs. In a clinical setting, NO is used in smaller quantities (20 to 45 ppm) as an inhalation gas for pulmonary hypertension [12]. In order to measure the exact NO concentration, several studies have mostly used Clark-type NO electrodes [56,57]. Such measurement of the NO concentration in the gel matrix of the IBIDI slides was not possible in this experimental setup. Furthermore, the objective of this study was to examine whether NO generally has an influence on the mentioned PMN functions, which is why a supratherapeutic NO concentration was aimed for in the tests. NO was examined as a dichotomous variable. Therefore, the NO concentration was not varied.

In the flow cytometric tests, the PMS buffer in the NO group was fumigated with 200 ppm NO using a gas-washing bottle. At room temperature (20 °C), according to Henry´s law, this corresponds to a concentration of approximately 3 nM [12,53,54].

### 4.5. Limitations of the Selected Method

Generally, in vitro laboratory tests are accompanied by the disadvantage of being unable to exactly reproduce the physiological conditions [12,53,54].

Density gradient centrifugation was used for PMN isolation. In this context, a study by Willnecker et al. was able to show a preactivation of PMNs by centrifugation, which, compared to microbead isolation, led to higher values of track length [58]. This fact should be taken into account when comparing PMN functionality from studies with different isolation methods. However, density gradient centrifugation is still a standard method and was, therefore, used for PMN isolation also in this study [59,60,61].

## 5. Conclusions

The studies conducted so far clearly support the controversially discussed topic of NO influence on PMNs. Our study, however, was able to demonstrate the migration-promoting effect of NO on PMNs. However, in the case of prior stimulation of PMNs, NO has no influence on the time component of their ROS production and NETosis functions. Moreover, the flow cytometer tests demonstrate an increase in ROS production in the case of pretreatment with NO.

The complex influence of NO on PMNs remains a central research topic since the exact mechanisms and other influencing factors are still not fully understood. Of particular interest is the examination of further factors that possibly modulate the effect of NO. What should be considered here are, among other things, the varying NO concentrations, as well as the NO influence before and after the activation of PMNs. Also, chemoattractant choice appears to influence the effect of NO on PMNs and thus represents a potential future research topic.

## Figures and Tables

**Figure 1 biomedicines-12-02353-f001:**
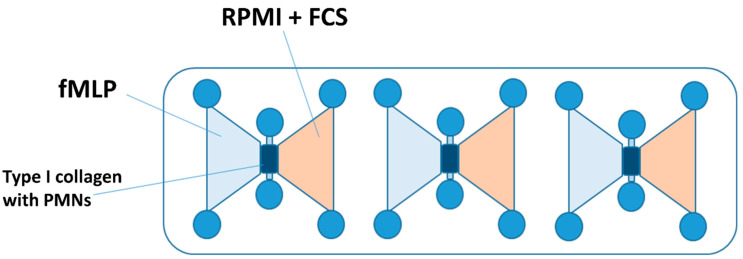
A schematic illustration of an IBIDI chemotaxis µSlide chamber. The gel matrix with the PMNs embedded was applied to the middle channel. The left reservoir was filled with the chemoattractant fMLP. The right reservoir contained the culture medium (RPMI + FCS), whereby a chemotaxis gradient was established. Abbreviations: PMNs = polymorphnuclear neutrophils; fMLP = N-formyl-L-methionyl-L-leucyl-phenylalanine; RPMI = Roswell Park Memorial Institute nutrient medium; and FCS = fetal calf serum.

**Figure 2 biomedicines-12-02353-f002:**
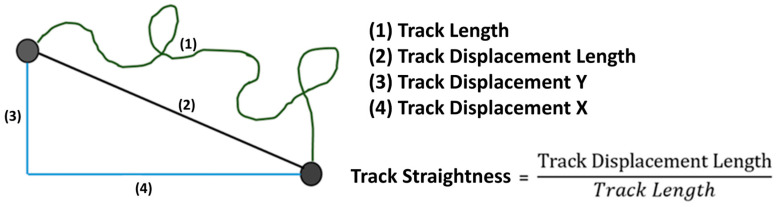
A graphical representation of the migration variables.

**Figure 3 biomedicines-12-02353-f003:**
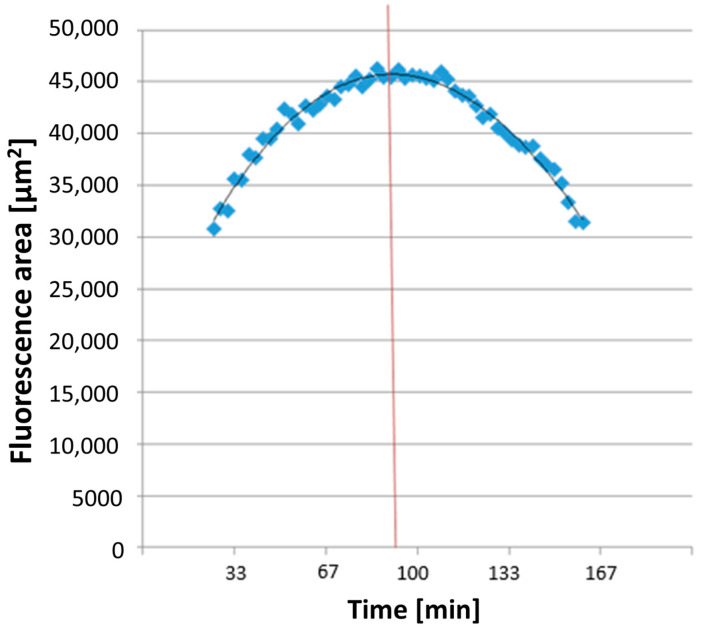
An exemplary illustration of TmaxROS analysis. The averaged line of the individual area sums corresponds to a function (see on top in the diagram), which was used to measure the maximum value (corresponding time extrapolated to the x-axis illustrated here as red line) of the area sums corresponding to the rhodamine fluorescence.

**Figure 4 biomedicines-12-02353-f004:**
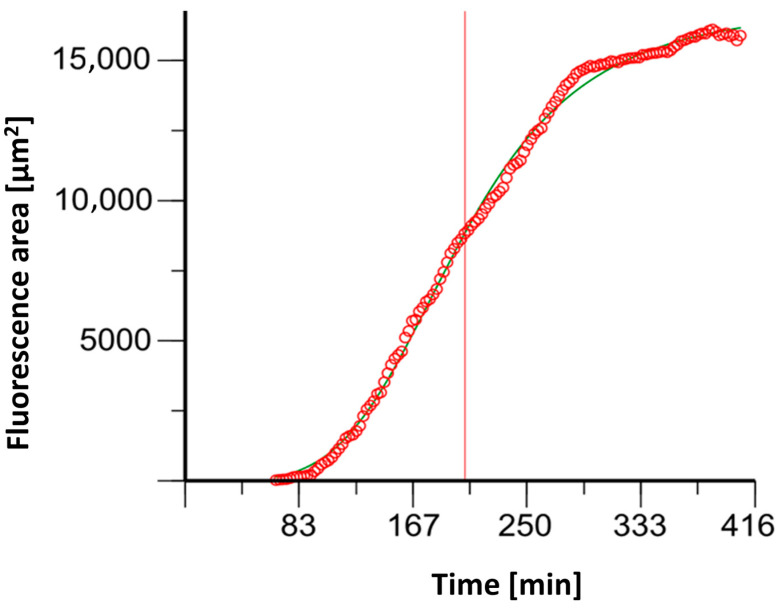
An example graph of the software Phoenix^®^ 8.0.0 to determine the ET_50_NETosis (corresponding time extrapolated to the x-axis illustrated here as red line).

**Figure 5 biomedicines-12-02353-f005:**
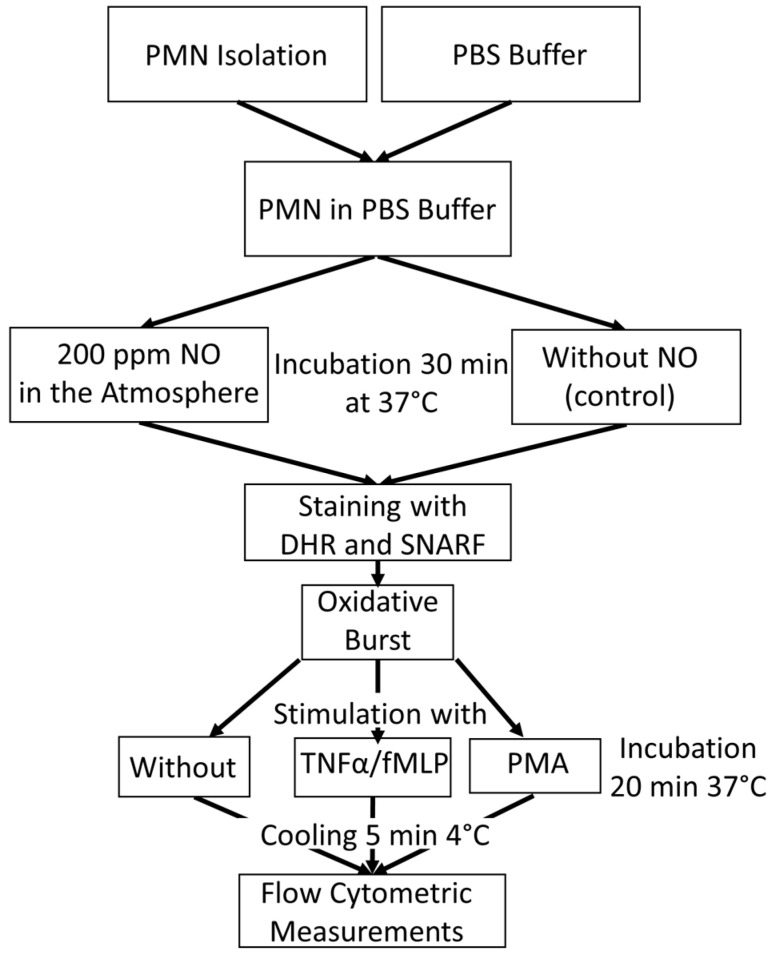
An overview of the experimental procedure for the tube preparation for oxidative burst measurement with flow cytometry.

**Figure 6 biomedicines-12-02353-f006:**
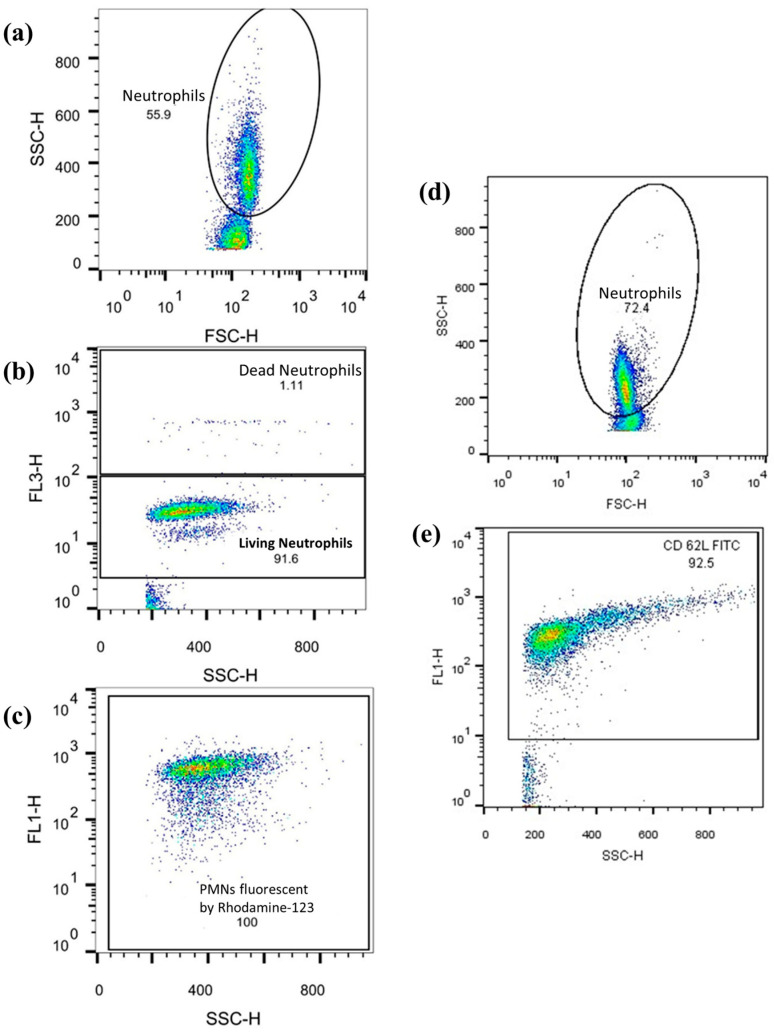
The example graphs of the FlowJo software: graphs (**a**–**c**) show the observation of ROS-producing PMNs. Numbers in the dot-plot area reflect the relative number [%] of the observed cell population. Axis description: SSC-H = side scatter; FSC-H = Front scatter; FL1 = Rhodamine or CD62L-FITC response; FL3-H = PI response. (**a**) PMNs were first isolated from other cell populations (lymphocytes, monocytes) according to size and granularity. (**b**) Vital cells were differentiated from dead cells. (**c**) The quantity of rhodamine fluorescence of the vital PMNs is shown. (**d**,**e**) Graphs show FITC fluorescence coupled to CD62L by anti-CD62L and again after PMN isolation.

**Figure 7 biomedicines-12-02353-f007:**
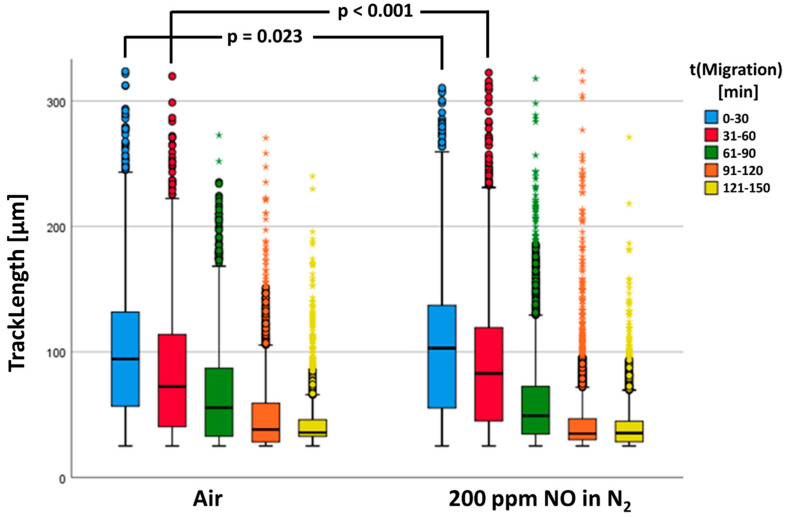
The track length is a significant difference between the two groups. The data are shown as a grouped box plot. The extreme values and outliers are shown in the form of circles and asterisks.

**Figure 8 biomedicines-12-02353-f008:**
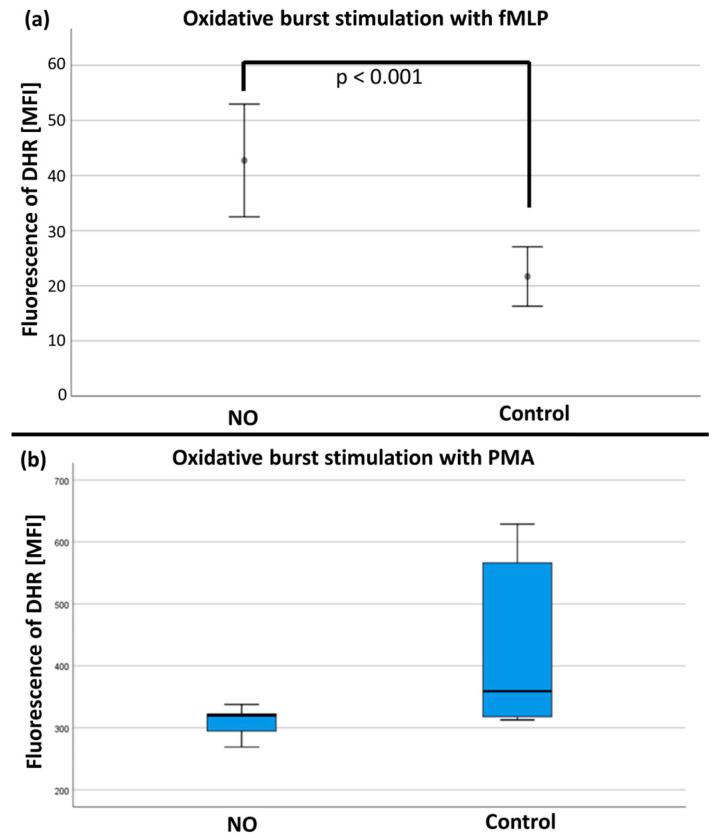
(**a**) The quantity of rhodamine fluorescence in MFI (mean fluorescence intensity) after fMLP stimulation with significant difference (*p* < 0.001) between the NO and control group. (**b**) The quantification of rhodamine fluorescence in MFI (mean fluorescence intensity) after PMA stimulation with no significant difference between the NO and control group.

**Table 1 biomedicines-12-02353-t001:** Burst experiment preparation for flow cytometric measurements. Incubation times are marked with x.

	Pre-Warmed PBS	Cell Suspension	DHR & SNARF	TNFα	37 °C 10 min	fMLP or PMA	37 °C 20 min	Samples on Ice
Basic Value	1 mL	20 µL	Each 10 µL				x	10 µL
TNFα + fMLP	1 mL	20 µL	Each 10 µL	10 µL		10 µL fMLP	x	10 µL
PMA	1 mL	20 µL	Each 10 µL			10 µL PMA	x	10 µL

**Table 2 biomedicines-12-02353-t002:** Antibody preparation for flow cytometric experiments.

	Isolated PMNs	Cold PBS Centrifuged	Remove Supernatant and Add:	Incubation 15 min 4 °C	Cold PBS Centrifuged Remove Supernatant	PBS
Basic Value	20 µL	1 mL			2 mL	200 µL
AntibodiesCD11b/CD62L/CD66b	20 µL	1 mL	5 µL		2 mL	200 µL

**Table 3 biomedicines-12-02353-t003:** The statistical evaluation of parameter TrackLength parameter for the first 60 min of the observation period.

TrackLength		Control (Air)	NO	*p*-Value
Time slot 0–30	*n*	3319	5449	0.023
	Median [µm] (IQR)	94.4 (75.1)	102.9 (103.1)	
Time slot 31–60	*n*	3052	6975	<0.001
	Median [µm] (IQR)	72.4 (73.4)	82.8 (74.4)	

**Table 4 biomedicines-12-02353-t004:** Statistical evaluation of parameter T_max_ROS and ET_50_NETosis.

TrackLength		Control (Air)	NO	*p*-Value
T_max_ROS	*n*	24	37	0.39
	Mean [min] (±SD)	111.1 (38.0)	86.3 (32.4)	
ET_50_NETosis	*n*	17	32	0.226
	Mean [min] (±SD)	214.3 (65.8)	235.6 (53.6)	

## Data Availability

The original contributions presented in the study are included in the article/Appendix A, further inquiries can be directed to the corresponding author.

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
