# Peer review of "Impact of Nitric Oxide on Polymorphonuclear Neutrophils’ Function"

_biomedicines, 2024, doi:10.3390/biomedicines12102353_

Round 1
Reviewer 1 Report
Comments and Suggestions for Authors
The authors have performed an interesting and original study. The relevance lies in expanding knowledge about neutrophils and their regulation. Neutrophils present the first line of non-specific immunity. These are very flexible cells that perform many functions. Their biochemistry and physiology of neutrophils are very complex and not fully understood. In the bloodstream, there is a pool of marginal neutrophils that move near the vascular wall. NO is also produced in this place. On the other hand, NO gas therapy is now actively used, and effects of high concentrations of NO on the immune system deserve study. I think the problem statement is adequate. The methods and approaches are chosen correctly. In general, the presentation of the results is good. However, while reading the article, I had questions that I hope the authors will take into account.
1. It seems to me that abbreviations should not be used in the title. I recommend replacing PMN with polymorphonuclear neutrophils
Abstract
2. The abstract, in my opinion, is unclear. Please rephrase the abstract more specifically. It is better to use the standard abstract outline: background, objective, methods, results, and conclusion.
3. In the abstract, please decipher fMLP.
Introduction
4. In the introduction, there is no need in breaking this section into six small subsections.
5. On line 44, (4, 5) are the references?
6. Please check all abbreviations to be deciphered, for example, LPS and HIV (line 51), MPO (line 83), PMA and fMLP (line 88), NET (line 83).
7. In the introduction, I think the part concerning the properties of NO could be shortened (line 19-70). In this section (references 1-15), use more recent sources.
8. I would also recommend briefly providing up-to-date information on the biochemistry and physiology of neutrophils, in particular their oxidative metabolism, chemotaxis, and NETosis.
9. The section on NO interactions with neutrophils could be expanded. For example, it should be noted that NO interacts with the superoxide anion radical to form toxic peroxynitrite, which leads to lipid peroxidation (DOI: 10.1016/s0891-5849(99)00250-6). It is also worth mentioning NO-regulated signaling, for example, DOI: 10.1016/j.bbrc.2010.12.123
Methods
10. In paragraph 2.2., indicate which anticoagulant was used to collect blood. Was it EDTA or heparin?
11. Also, in paragraph 2.2., indicate how many volunteers there were in the study. Neutrophils are known to be sensitive to any processes in the body, changing their activity during inflammation, allergies, etc. Provide criteria for inclusion and exclusion of volunteers in the study. How was it verified that the volunteers were healthy?
12. In the caption under Figure 1, provide an explanation of all the abbreviations used.
13. It seems to me that the microscope magnification should be indicated as 100× instead of ×100 (line 136).
14. Please decipher DAPI (4′,6-diamidino-2-phenylindole) in line 171.
15. Subsection 2.7. Why do you abbreviate “flow cytometry experiments” as FACS? It seems to me that FACS should stand for “fluorescence activated cell sorting”.
16. “2.7.1. Preparing the ROS measurement series” is the only subsection, there is no point in separating it.
17. What was the concentration of NO in the buffer solution and after mixing with neutrophils? This information is in the Discussion and Limitations section and should be presented here. A rationale for the choice of concentration should also be given - move this information from the Limitations section to the Discussion.
18. Line 188 - what concentrations of PMA and fMLP were used?
19. Was PMA used later in the study? I did not find these experiments in the text.
20. Line 192 - what does SNARF stand for?
21. Line 192 - give the concentrations of rhodamine and SNARF.
22. Line 193 - Give the concentration of TNFalpha, not just the volume.
23. Table 1 - Replace the volumes with the final concentrations.
24. The same for Table 2. Instead of volumes, provide the final concentrations.
25. The information in subsection 2.7.1 is rather difficult to perceive. I recommend providing a graphical scheme of the experiments (the experimental design).
26. In paragraph 2.10, provide the critical p-values at which the differences were considered significant.
Results
27. In Table 3, provide the interquartile range along with the medians.
28. In paragraph 3.2., I believe that numerical data should be provided as a table, not just the p-values.
29. In Fig. 7b, NO treatment leads to a significant decrease in data variability compared to the control. Is there an explanation for this fact? Is it worth paying attention to?
Discussion
30. The conclusion in subparagraph 4.1 is unclear. Please formulate it. Subparagraph 4.1. seems unfinished.
31. Line 294 – say a few words about the studied epitopes CD 11b, CD62L, CD66b.
32. In the title 4.3. the term “oxidative stress” is not quite correct. The title should sound like, for example,” Influence of NO on oxidative metabolism” or “ROS metabolism”
33. Line 331. I am confused by the phrase: ”The results showed that NADPH oxidase caused a significant inhibition of ROS production when incubated for 5 minutes with NO before the activation”. If you mean that NO treatment inhibited NADPH oxidase activity, the phrase needs to be reformulated.
34. It seems to me that in subsection 4.3. it is necessary to indicate which ROS were determined. Neutrophils produce superoxide anion radical, hydrogen peroxide and hypochlorite (these are the main ROS). It is necessary to indicate which ROS dihydrorhodamine 123 is sensitive to (is it a superoxide anion radical?). And in the discussion below, indicate which fluorescent probes other researchers used to determine ROS. Differences in the results may be due, among other things, to different methods for assessing the level of ROS.
35. The activity of neutrophils is also affected by the isolation procedure. Density gradient centrifugation, for example, activates neutrophils. This point should also be addressed in the discussion.
36. In paragraph 4.5, most of the information should be moved to the Introduction, Materials and Methods, and Discussion. This section should be significantly shortened to a few sentences.
Reviewer 2 Report
Comments and Suggestions for Authors
1. The writing structures appeared to be strange. For example, there should be no need to divide introduction and discussion into different sections. Also, the methods should be put at the end of manuscript. But this is indeed an issue for the editor.
2. In the abstract, the authors did not write the results about surface epitopes.
3. How many times for the experiments have been repeated?
4. Did the authors use freshly isolated cells, or they cultured the cells and used the cells after several passages? Did they pool the cells from different volunteers?
5. How about the measurement of intracellular NO? Since the authors measured ROS, it should be possible for them to measure intracellular NO.
Round 2
Reviewer 1 Report
Comments and Suggestions for Authors
The authors have done a great job and have taken into account all my comments. I am completely satisfied. The article can be accepted for publication.
Reviewer 2 Report
Comments and Suggestions for Authors
All the comments have been addressed.